# Twenty-Five Years of Propagation in Suspension Cell Culture Results in Substantial Alterations of the *Arabidopsis Thaliana* Genome

**DOI:** 10.3390/genes10090671

**Published:** 2019-09-02

**Authors:** Boas Pucker, Christian Rückert, Ralf Stracke, Prisca Viehöver, Jörn Kalinowski, Bernd Weisshaar

**Affiliations:** 1Genetics and Genomics of Plants, Faculty of Biology, Center for Biotechnology (CeBiTec), Bielefeld University, Sequenz 1, 33615 Bielefeld, NRW, Germany (R.S.) (P.V.) (B.W.); 2Microbial Genomics and Biotechnology, Center for Biotechnology (CeBiTec), Bielefeld University, Sequenz 1, 33615 Bielefeld, NRW, Germany (C.R.) (J.K.)

**Keywords:** copy number variations, variant calling, next generation sequencing, long read sequencing

## Abstract

*Arabidopsis thaliana* is one of the best studied plant model organisms. Besides cultivation in greenhouses, cells of this plant can also be propagated in suspension cell culture. At7 is one such cell line that was established about 25 years ago. Here, we report the sequencing and the analysis of the At7 genome. Large scale duplications and deletions compared to the Columbia-0 (Col-0) reference sequence were detected. The number of deletions exceeds the number of insertions, thus indicating that a haploid genome size reduction is ongoing. Patterns of small sequence variants differ from the ones observed between *A. thaliana* accessions, e.g., the number of single nucleotide variants matches the number of insertions/deletions. RNA-Seq analysis reveals that disrupted alleles are less frequent in the transcriptome than the native ones.

## 1. Introduction

*Arabidopsis thaliana* is a small flowering plant distributed over the northern hemisphere and has become the model system of choice for research in plant biology. In 2000, the genome sequence was released as the first available plant genome sequence [1]. After generating this reference sequence from the accession Columbia-0 (Col-0), many other *A. thaliana* accessions were analyzed by sequencing to investigate, among many other topics, genomic diversity, local adaptation, and the phylogenetic history of this species [2,3,4]. While most initial re-sequencing projects relied on short read mapping against the Col-0 reference sequence [5,6,7], technological progress enabled de novo genome assemblies [2,8] which reached a chromosome-level quality [9,10,11]. Low coverage nanopore sequencing was also applied to search for genomic differences, e.g., active transposable elements (TEs) [12].

Due to the high value of *A. thaliana* for basic plant biology research, it is frequently grown in greenhouses under controlled and optimized conditions. Previous studies investigated the mutation rates within a single generation [6,13]. Mutational changes appear to be different between plants grown under controlled conditions and natural samples collected in the environment [6]. Another approach harnessed an *A. thaliana* population in the United States of America, which is assumed to originate from a single ancestor thus showing mutations accumulated over the last decades [14]. This study investigated modern and ancient specimens and estimated a rate of 7.1 × 10^−9^ substitutions per site per generation [14]. Since not all mutations are fixed during evolution, the mutation rate is higher than the substitution rate.

Even further away from natural conditions in the environment is the propagation of cells in suspension cultures. Cells from such cultures can easily be employed for transient transfection experiments [15]. Transient transfections of At7 protoplasts are a relatively straightforward method to study promoter structure and activity and to investigate the interactions between transcription factors and promoters of putative target genes [16,17].

Since most functions of plants are dispensable in suspension culture, it was expected that mutations in these dispensable genes accumulate over time due to genetic drift or even due to positive selection. We sequenced and analyzed the genome of At7 cells which have been propagated in suspension culture for about 25 years and identified massive genomic changes.

## 2. Methods

### 2.1. Plant Material

The *A. thaliana* suspension cell culture At7 [18] is derived from hypocotyl of the reference accession Columbia (Col). This cell culture is cultivated at 26 °C and 105 rpm in darkness with sugar supply on a rotary shaker. A subset of cells is transferred into fresh B5 medium at a weekly basis. For details about the propagation of this cell culture, see [19]. All sequencing data sets generated in this study were submitted to the European Nucleotide Archive (ENA) as part of the study PRJEB33589 and the sample ERS3588070.

### 2.2. RNA Extraction and RNA-Seq

Total RNA was extracted based on a previously described protocol [16]. Based on 1 µg of total RNA, sequencing libraries were constructed following the TrueSeq v2 protocol. Three biological replicates of the At7 cell culture (splitted the preceding week) were processed and combined for sequencing after one week of incubation. Single end sequencing of 83 nt was performed on an Illumina NextSeq500 at the Sequencing Core Facility of the Center for Biotechnology (CeBiTec) at Bielefeld University (ERR3444576, ERR3444577). 

### 2.3. DNA Extraction

Cells were separated from media through filtration on a Büchner funnel. Next, cell walls were destroyed by treatment in the Precellys Evolution (QIAGEN, Hilden, Germany) for 3 × 30 s at 8500 rpm with 1 mm zirconium beads (Roth). The following DNA extraction was based on a previously described cetyltrimethylammoniumbromid (CTAB) protocol [20]. To increase the purity of the DNA an additional ethanol precipitation step was added. DNA from the same extraction was used for sequencing with different second and third generation technologies.

### 2.4. Oxford Nanopore Sequencing

DNA was quantified via a Qubit fluorometer (Invitrogen, a brand of Thermo Fisher Scientific, Waltham, MA, USA) following suppliers’ recommendation. Sequencing was performed on a total of four flow cells. The Ligation Sequencing Kit SQK-LSK109 (Oxford Nanopore Technologies, ONT, Oxford, UK), and the Rapid Sequencing Kit SQK-RAD004 (ONT) were used to prepare two DNA libraries each for sequencing based on the suppliers’ protocol. Sequencing on a MinION (ONT) (ERR3445571) and GridION (ONT) (ERR3445572), respectively, was performed using R9.4 flow cells (ONT) based on the supplier’s instructions. Real time base calling for the MinION experiments was performed on a MinIT (ONT) via MinKNOW software (ONT).

### 2.5. Illumina DNA Sequencing

Sequencing library preparation was performed as described previously [8] and sequencing took place at the Sequencing Core Facility of the CeBiTec at Bielefeld University. Paired-end sequencing of the library was done on an Illumina HiSeq1500 system in Rapid Mode to produce 2 × 250 nt reads (ERR3445426) and on an Illumina NextSeq500 benchtop sequencer resulting in 2 × 155 nt reads (ERR3445427).

### 2.6. Coverage Analysis

Dedicated Python scripts were applied for the genome-wide coverage analysis, the classification of genes based on coverage values, and the investigation of coverage values at variant positions. These scripts are available at Github: https://github.com/bpucker/At7. Completely deleted regions were identified through the identification of zero coverage regions in At7, i.e., regions where no At7 reads are mapped to the Col-0 reference sequence, as described before [8].

### 2.7. Genome Assembly

ONT reads longer than 3 kb were subjected to genome assembly via Canu v1.8 assembler [21] using the following parameters: ‘genomeSize=145m’ ‘useGrid=1’ ‘saveReads=true’ ‘corMhapFilterThreshold=0.0000000002’ ‘ovlMerThreshold=500’ ‘corMhapOptions=--threshold 0.80 --num-hashes 512 --num-min-matches 3 --ordered-sketch-size 1000 --ordered-kmer-size 14 --min-olap-length 2000 --repeat-idf-scale 50’. In addition to Canu, Miniasm v0.3-r179 [22] and Flye v2.3.1 [23] were run to identify the best assembler for this data set as previously described [11]. The quality of assembled contigs was improved through nine successive rounds of polishing with Nanopolish v0.11 [24] harnessing the information of all ONT reads. Minimap2 v2.10-r761 [22] was used for the read mapping with the arguments ‘--secondary=no -ax map-ont’. Next, Illumina reads were mapped via BWA MEM v0.7.13 [25] to the polished assembly. Five successive polishing rounds with Pilon v1.22 [26] were performed based on Illumina read mappings. Previously developed Python scripts [8] were applied to check the assembly for contaminations, to remove very short contigs, and to calculate general assembly statistics.

### 2.8. Variant Calling and Annotation

Illumina sequencing reads of At7 were aligned to the TAIR9 reference sequence of Col-0 via BWA MEM v0.7.13 [25]. Next, GATK v.3.8 [27,28] was applied for the identification of small sequence variants as previously described [29]. In contrast to previous studies, variant positions with multiple different alleles were kept as they are biologically possible. GATK is able to identify low frequency variants, thus providing a powerful solution given the variation in ploidy between hemizygous and pentaploid. Variants were considered as heterozygous if at least 5 reads and at least 5% of all reads support an additional allele besides the dominant one. Additionally, variants were excluded if their coverage deviates from the average coverage of a 2 kb flanking region by more than 20%. SnpEff [30] was deployed to assign predictions of the functional impact to all small sequence variants. Previously developed Python scripts [8,29] were customized to investigate the distribution of variants and to check for patterns. SVIM [31] was deployed to identify large variants based on a Minimap2 mapping of all ONT reads against the Col-0 reference sequence. RNA-Seq reads were mapped via STAR [32] to the Col-0 reference genome sequence using previously described parameters [33]. Samtools [34] and bedtools [35] were used to call variants based on an RNA-Seq read mapping to the Col-0 reference sequence.

## 3. Results and Discussion

### 3.1. Ploidy Differences Between At7 and Col-0

Mapping of the At7 Illumina short reads and ONT long reads to the Col-0 reference sequence was used to assess genomic changes in At7. The coverage analysis revealed duplications and deletions of large chromosomal segments (Figure 1, Appendix A). Similar regional variations in ploidy between neighbouring chromosomal segments are common in immortalized insect and mammalian cell lines and tumors, where they may be an advantage to cells [36,37]. In the At7 suspension cell culture, about 5 Mbp at the northern end of chromosome 2 (Chr2) and chromosome 4 (Chr4) appear highly fragmented. In addition, regions around the centromeres are apparently fragmented, but this could be an artifact of higher repeat content and a substantial proportion of collapsed peri-centromeric and centromeric sequences. These differences in chromosomal stability seem to be consistent between plants and animals since similar observations were also reported, e.g., Chinese hamster ovary (CHO) and *Drosophila* cell lines [36,37].

An average coverage of 50-fold in a small region at 1 Mbp on Chr1 and in a region between 9.5 Mbp and 11 Mbp on Chr2 indicates hemizygosity, i.e., there is only one copy left and the regions are hemizygous. Small blocks in the northern end of Chr4 show a similar coverage. An average coverage of 150-fold indicates that most other regions are triploid. Although the starting material from Col-0 was diploid 25 years ago, these regions represent a minority in At7. A region on Chr1 (17.5 to 21 Mbp) is present in five copies (pentaploid). There are only 116 regions larger than 100 bp which appeared of being completely deleted in At7. These observations in At7 differ from findings in CHO cells, where large chromosomal segments were found in a hemizygous state [37]. Possible explanations for the observed polyploidy regions are defective mitosis and chromosome endoreplication, as previously reviewed by D’Amato [38] and references therein. Similar chromosomal restructuring has been reported during genome elimination in an *A. thaliana* plant with different centromer-specific H3 histones [39].

Besides Illumina sequencing, Oxford Nanopore Technologies’ MinION and GridION sequencer were deployed to generate long reads for the detection of large structural variants. Alignments against the Col-0 reference sequence revealed 160,348 deletions and 5902 insertions (Appendix A). Previous comparisons of natural *A. thaliana* accessions revealed equal numbers of insertions and deletions [2,8,29]. For comparison, long reads of the Nd-1 accession were aligned to the Col-0 reference sequence resulting in the identification of almost equal numbers of insertions and deletions. Although different long read sequencing technologies are involved, this substantial difference between the At7 comparison to Col-0 and the Nd-1 comparison to Col-0 is unlikely to be a technical artifact. Analysis of the remaining coverage in deletions supports the validity because most of these regions seem to be maintained in at least two haplophases (Appendix A). The excess of deletions observed here indicates that a single haplophase of the At7 genome is probably smaller than a haploid Col-0 genome. This could be due to an ongoing genome size reduction in the At7 cell culture. A subset of the At7 cells in the suspension culture is transferred to a new flask with fresh media on a weekly basis. Fast-dividing cells have a higher chance of passing at least one daughter cell to the new flask. The result is a strong selection pressure for fast cell division. Although the replication of additional sequences is probably not costly in terms of evolution, the transcription and translation of dispensable genes was shown to be under negative selection in eukaryotes [40]. Artificial selection through humans is one of the strongest forces that can drive evolutionary changes [5,6,14]. The comparison of the genome sequences of *A. thaliana* and the close relative *A. lyrata* revealed randomly distributed small deletions as one major source for the genome size differences [41]. High numbers of large deletions between At7 and Col-0 indicate that the haploid genome size reduction is occurring at substantially higher speed and using different mechanisms.

Read mappings revealed that both, plastome and chondrome, are still detectable in At7 cells. Due to differences in the cultivation conditions and DNA extraction methods, a reliable comparison of plastome coverage to nucleom coverage between At7 and plants is not feasible. Although the loss of chloroplasts may seem likely in plant cells incubated in the dark and with sugar supply through the cultivation media, several central biosynthetic pathways like de novo synthesis of fatty acids [42,43] and isoprenoid biosynthesis [44] are located in the plastids [45]. This is probably the explanation why this organelle is maintained in At7 cells. The coverage ratio between chondrome and plastome is about 10 times higher than observed in native plants [11,46] (Appendix A). The increased number of mitochondria could be due to the specific conditions cultured At7 cells are exposed to. Previous reports described differences in organelle numbers due to development and environmental stresses [45].

### 3.2. Sequence Variants and Copy Number Variations

Mapping of all At7 Illumina sequencing reads against the Col-0 reference sequence and following variant calling revealed an almost equal amount of single nucleotide variants (SNVs) and small insertions/deletions (InDels) (Appendix A). This is in contrast to previous re-sequencing studies where natural populations displayed a substantial excess of SNVs compared to InDels [2,5,8]. In total, 127,686 small variants were identified leading to a frequency of about one variant per kbp. This frequency is lower than previously reported values of 1 variant in 200–400 bp for the comparison of natural *A. thaliana* accessions [5,8]. The sequence variants show an increased change from CG to AT, when compared to the Nd-1 vs. Col-0 comparison [8,29]. Variants in At7 are the result of spontaneous mutations which accumulate in the asexually reproducing At7 cells according to Muller’s Ratchet [47] as previously reported for apomicts [48,49]. The higher ratio of InDels could indicate that more variants are deleterious than usually found between sexually reproducing accessions. This would be in agreement with the expectation based on Muller’s Ratchet. A generally higher frequency of variants between accessions could be due to rare outcrossing events, which can introduce numerous variants in a single event [50]. It is even possible that several of these variants between accessions compensate each other’s effect [2,29], while isolated variants in At7 are deleterious. In agreement with previous findings [5,8], the genome-wide distribution of variants between At7 and the Col-0 reference sequence showed increased variant frequencies around the centromeres (Figure 2). As previously suggested [8], this enrichment can be partly explained by a higher proportion of collapsed sequences in these regions due to the higher abundance of repeats.

About 94.2 % of all small sequence variants display deviating frequencies. This can be explained by the asexual reproduction of the originally diploid At7 cells. A mutation only occurs in one allele and is consequently passed on to all daughter cells. However, the other allele of the same locus would still show the original sequence. It is very unlikely that the same mutation occurs in all alleles independently. Although exchange between alleles cannot be excluded, it is not a relevant mechanism for the generation of homozygous variants. Many homozygous variants between At7 and Col-0 are clustered in a few regions (Figure 2). In most cases, the respective regions are also depleted of variants with deviating frequencies. One clear example of a homozygous variant stretch is located on Chr2 between 9.5 Mbp and 11 Mbp (Figure 2, magenta marked) where the number of homozygous variants exceeds the number of variants with deviating frequencies. These 2774 homozygous variants were probably caused by the hemizygosity of these regions in the At7 genome (Figure 1). A substantially reduced read coverage of homozygous variants compared to variants with deviating frequencies supports this hypothesis (Figure 3) and regions with high proportions of homozygous variants displayed generally low coverage. A small fraction of variants with deviating frequencies could be due to spurious read mappings and thus false positive variant calls. The weak peak of variants with about 2-fold Illumina coverage but only one detectable allele might be explained by duplication of the respective genome region after the small variant mutation occurred. The observation of a few variants with deviating frequencies in apparently hemizygous regions could be due to spontaneous mutations in some cells leading to different lineages. However, genetic drift will reduce this diversity resulting in the dominance of one allele as observed for most positions and in agreement with Muller’s Ratchet [47].

A minority of the homozygous variants could also have been contributed by the original material which was used to generate this suspension cell cultures. Based on previous studies of mutation accumulation over generations [5,6,8,14], it is likely that a few of these variants have been differentiating the material used for the generation of the At7 cell culture from the material used in the *A. thaliana* genome sequencing project. Errors in the At7 sequencing data and in the Col-0 reference sequence [7,11,51] could be an additional source of seemingly differences. Similar numbers were observed between genomes of individual plants of the same *A. thaliana* accessions before and have been considered to be recent events or technical artifacts [8,9].

Since extremely large differences in the genome structure of At7 and Col-0 might not be revealed through variant detection tools, genes of the Col-0 genome sequence annotation Araport11 were classified based on their read mapping coverage (Figure 4, Appendix A). About 85% of all genes display a coverage which was similar to the coverage of flanking genes. Therefore, it is likely that large genomic blocks, and not just single genes, were deleted or duplicated. This is in agreement with consistent coverage levels for large genomic blocks. Again, an average coverage of 50-fold indicates hemizygous regions, while multiples of this hemizygous coverage value indicate the presence of additional copies. The high number of genes with increased copy numbers is in agreement with previous reports of increases in chromosome duplications over cultivation time as callus [52]. Analysis of the genome of this cell line again after additional years of cultivation could reveal if there is a saturation of this increase in ploidy and nuclear DNA content. We speculate that these copy numbers are beneficial due to deleterious variants in some copies of required genes.

### 3.3. Variant Impact on Genes and Transcript Isoform Abundance

The impact of InDels on coding sequences was analyzed by comparing the length distributions of InDels inside protein encoding sequences with the InDel length distribution outside of these coding regions. InDel lengths which are divisible by three (one codon size) are enriched inside of coding sequences. Of all 28,213 analyzed InDels, only 856 are located within the 33.5 Mbp of protein encoding regions annotated in Araport11. This depletion of InDels inside of protein encoding regions indicates that at least residual selection against disruption of these sequences is still ongoing.

Assessment of the functional impact of small sequence variants revealed a high impact effect (e.g., premature stop codon or frameshift) on a total of 2189 genes (Appendix A). This high number can be explained by functional redundancy due to multiple alleles, i.e., at least one allele is maintained in a functional state. In addition, many genes might be dispensable under stable, stress-free cell culture conditions. Therefore, the accumulation of disruptive variants or entire deletion is feasible. We restrained from gene ontology (GO) enrichment analysis due to a functional high redundancy caused by the presence of multiple alleles for most genes.

RNA-Seq analysis revealed a substantial difference in the abundance of native alleles and defective alleles. Isoforms with a destructive variant displayed a substantially reduced abundance (Figure 5, Mann-Whitney U-test *p*-value = 0). The genomic coverage of these variant positions displays a very similar pattern: Abundance of the reference allele is on average substantially higher than the coverage of the alternative alleles. This indicates that not the nonsense mediated decay pathway [53], but different copy numbers of the allele, are the main explanation for the difference in transcript abundance. Such copy number-dependent preferential allele expression is known from the vegetatively propagated autotetraploid potato [54] and seems to be a general mechanism.

### 3.4. De Novo Genome Assembly

To reveal the sequences of large structural variants, we generated an independent de novo genome assembly of At7. The Canu assembler performed best in our hands. A set of 1.4 million ONT reads longer than 3 kb was assembled into 433 contigs representing 126.2 Mbp of the At7 nucleom (Appendix A). The N50 of 1.2 Mbp is substantially lower than other recent reports of *A. thaliana* genome assemblies [9,10,11]. We speculate that points with abrupt changes in coverage of the At7 genome deteriorated the assembly contiguity. Manual inspection of the borders of some hemizygous regions revealed two groups of reads which support different assembly paths: one containing the hemizygous sequence and one continuing in a different genomic location. The observation of different groups of reads is in agreement with the high number of large structural variants, which are not supported by all reads, and several distinct coverage levels detected during the read mapping to the Col-0 reference sequence. However, the total assembly size exceeds the 119.8 Mbp of the Col-0 reference sequence thus rendering the accurate anchoring of contigs in some regions, e.g., close to the centromeres, unreliable.

### 3.5. Future Directions

After the identification of genomic changes in At7 compared to the Col-0 reference genome sequence, it would be interesting to investigate genomic changes in other cell culture lines. A previous study indicates that the speed of genomic changes could be cell line specific [55]. Based on our findings and previous reports about differences in genomic stability [37], there might be certain regions in the Arabidopsis genome which are more likely to change than others as previously reported for *Solanum tuberosum* [56]. Aneuploidy events were described previously for an asexually reproducing plant [48] and in long term suspension and callus cultures [57,58]. Comparing genomic changes observed in independent lineages would facilitate the identification of instable or even dispensable regions. Increased copy numbers of many genomic regions seemed to have low metabolic costs, but might be beneficial in buffering deleterious mutations that accumulate due to asexual reproduction.

In this study, we focused at the genome sequence and excluded other previously reviewed DNA modifications like methylation and chromatin condensation [59,60]. The methylation pattern is known to partly change, e.g., in response to environmental conditions or developmental stage [61,62]. Changes in methylation are associated with transposable element activity [63] and general genome instability [56]. TE activation in At7 could be harnessed to identify intact elements, which are repressed by methylation under normal conditions in a plant. Local hypo- and hypermethylation were reported before as the result of a tissue culture and appeared at the same genomic location across independent lines [64,65]. Generally increased methylation levels have been reported to render in vitro cultures unsuitable for long term production of secondary metabolites [66]. In the light of these reports and our genomic alteration findings, substantial methylation differences between At7 cells in a suspension culture and hypocotyl cells in a Col-0 plant can be expected. Since the number of hypocotyl cells in a plant is very limited, whole genome nanopore sequencing of this cell type to generate a reference data set for the methylation pattern is not feasible at the moment. Advances in single cell sequencing could enable such a comparison in the future.

In-depth investigation of genomic changes under cell cultures conditions including methylation differences could benefit applications like the development of stable transgenic lines from in vitro cell cultures [67]. Using regenerable protoplasts for non-transgenic genome editing could advance crop improvements [56] and plant cells could even be used as efficient biofactories once epigenetic challenges are overcome [66].

## Figures and Tables

**Figure 1 genes-10-00671-f001:**
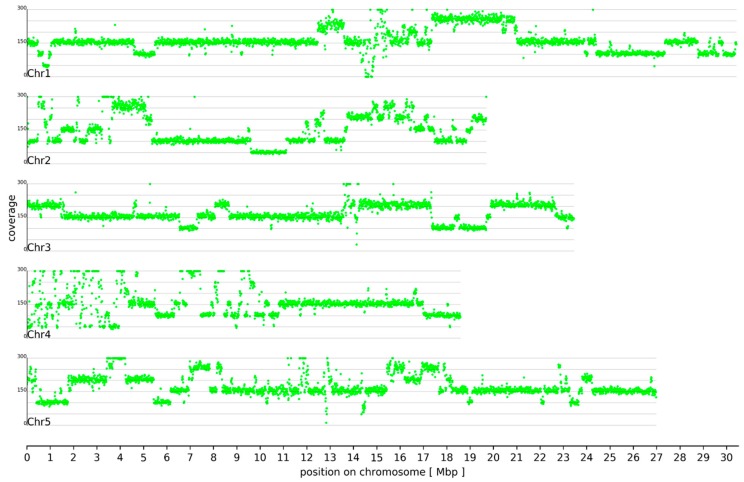
Genome-wide coverage of the Columbia-0 (Col-0) reference sequence. Hemizygous regions in At7 were revealed by a read coverage of approximately 50-fold when combining Illumina and ONT sequencing reads. Different multiples of this values can be observed, revealing the presence of large scale multiplications.

**Figure 2 genes-10-00671-f002:**
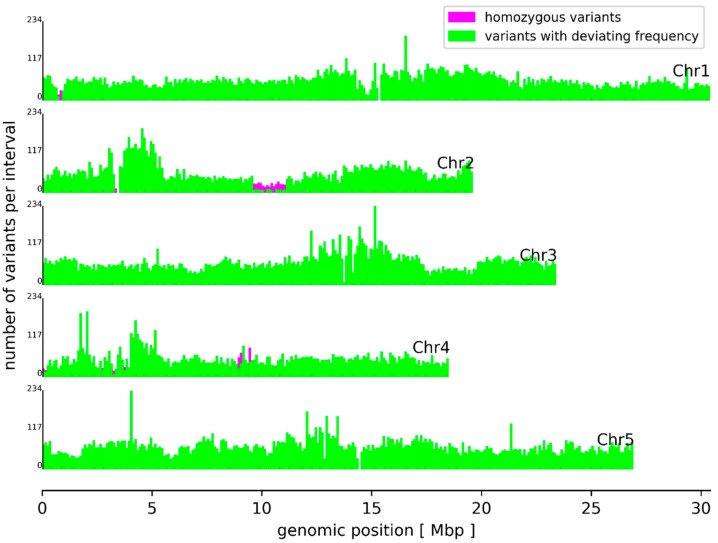
Genome-wide distribution of small sequence variants between At7 and Col-0. Homozygous variants (magenta) and variants with deviating frequency were counted in genomic blocks of 100 kb on all five chromosomes of the Col-0 reference sequence.

**Figure 3 genes-10-00671-f003:**
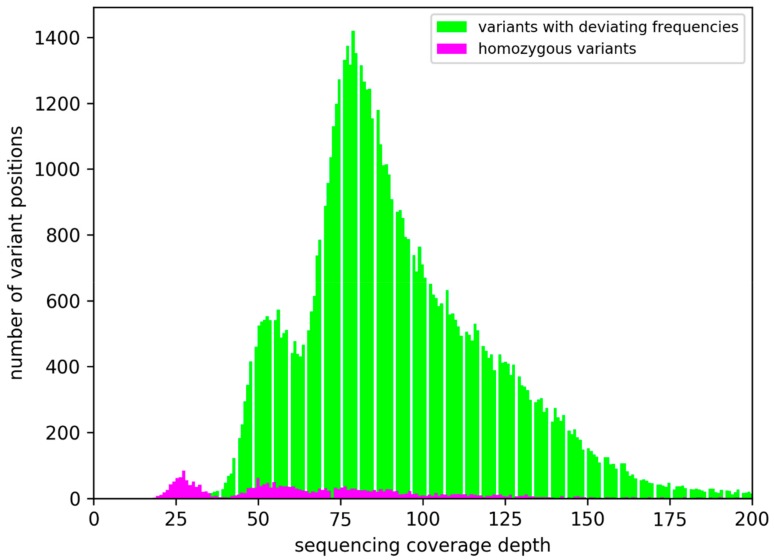
Illumina sequencing coverage depth at variant positions. Only Illumina reads were considered in this analysis, because the variant calling was limited to this set of high quality sequences. Therefore, the average coverage of 25-fold is about half the coverage observed for the complete sequence read data set.

**Figure 4 genes-10-00671-f004:**
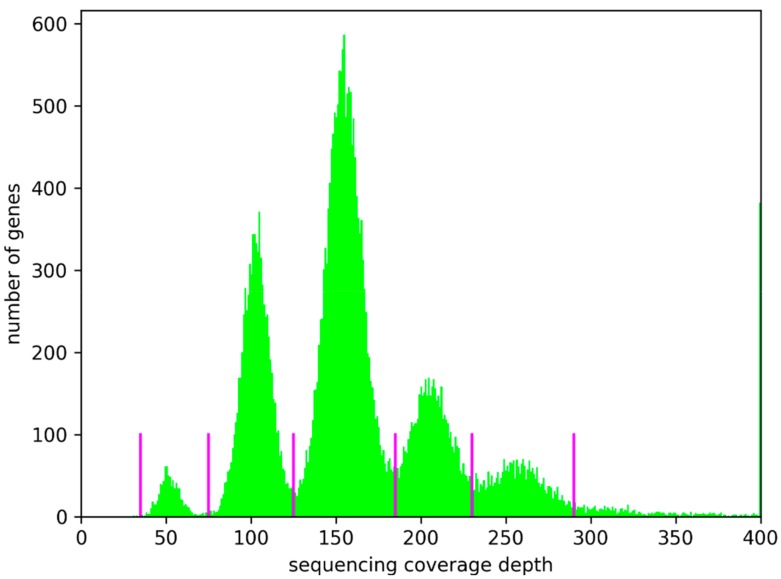
Gene copy numbers. Read coverage per gene was calculated as proxy for the copy number of the respective gene. Magenta lines indicate the central position of coverage valleys enclosed by peaks. Distances between these coverage valleys are not identical due to differences in peak height and resulting width differences. Absolute coverage values might be underestimated due to the removal of read pairs, which appeared as the result of PCR duplicates. Values above 400 are included in the largest bin to allow accommodation in one figure.

**Figure 5 genes-10-00671-f005:**
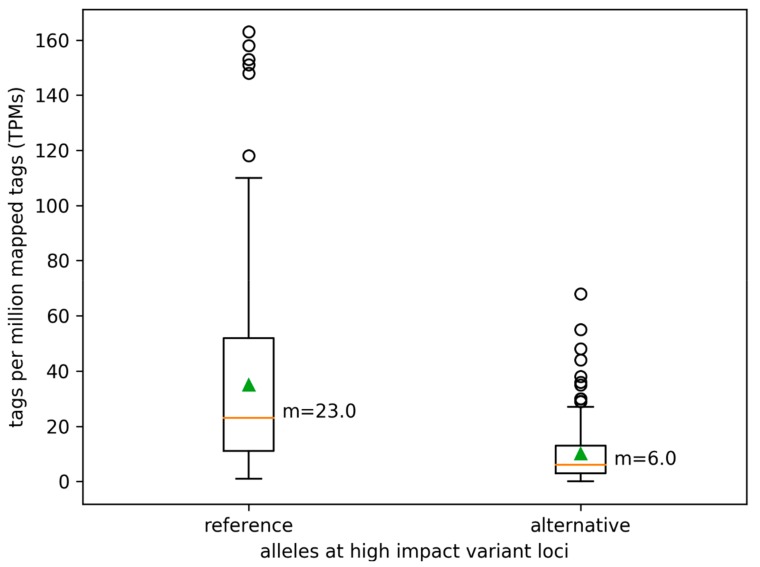
Allele specific transcript abundance at high impact variant positions. The number of RNA-Seq reads supporting the reference (Col-0) and alternative allele, respectively, were determined at high impact variant positions.

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
