# Peer review of "Twenty-Five Years of Propagation in Suspension Cell Culture Results in Substantial Alterations of the Arabidopsis Thaliana Genome"

_genes, 2019, doi:10.3390/genes10090671_

Round 1

Reviewer 1 Report

This study is based on sound experimental work, which is professionally presented.

Indeed, the manuscript presents data which may be of interest for the plant scientific community.

I have just few issues which should be considered to strengthen the quality of the manuscript.

- The results section is split up in too many sub-sections, some of them can be merged (e.g. 3.4/3.5, 3.1/3.3).

- The manuscript lacks the discussion section. In the “results and discussion” the Authors actually present only the results. There are several points that could be examined in this section: an evolutionary point of view, or a comparative analysis with other genomic studies which are cited by the Authors, and also a methodological evaluation on the use of cell cultures considering these results.

Author Response

Dear Reviewer,

Thank you very much for your time, efforts, and constructive suggestions. Please see below a detailed response to your points.

Best wishes on behalf of all authors,
Boas Pucker

1) The results section is split up in too many sub-sections, some of them can be merged (e.g. 3.4/3.5, 3.1/3.3).

We combined the sub-sections to reduce the number of sub-sections to five:
3.1 Ploidy differences between At7 and Col-0 (former sub-sections 3.1 + 3.3 + 3.8)
3.2 Sequence variants and copy number variations (former sub-sections 3.2 + 3.4)
3.3 Variant impact on genes and transcript isoform abundance (former sub-sections 3.5+3.7)
3.4 De novo genome assembly (former sub-sections 3.5)
3.5 Future directions (new sub-section to extend discussion of findings)

2) The manuscript lacks the discussion section. In the “results and discussion” the Authors actually present only the results. There are several points that could be examined in this section: an evolutionary point of view, or a comparative analysis with other genomic studies which are cited by the Authors, and also a methodological evaluation on the use of cell cultures considering these results.

We expanded our discussion substantially by adding comparisons to previous studies, evolutionary implications, and a section about future steps. Due to the complex nature of these changes, we do not provide specific line numbers, but point to the whole extensively revised "results&discussion" section.

Reviewer 2 Report

Summary

Pucker et al applied different sequencing technologies for short reads RNAs and long reads of DNAs in order to study the genome structure of the At7 cell culture which has been kept for 25 years. Pucker et al has demonstrated striking differences in the genomes of At7 and Columbia-0 (Col-0) despite that At7 was originally derived from the hypocotyls of Col-0. This study is a continuation of Pucker et al 2016 [1] to study the pan-genome of A. thaliana. The current study’s results were informative and already in bioRxiv [2] as preprint. However, to assist the Genes readers in understanding the Pucker et al’s methodologies and results, it is advised to introduce full names and apply conventionally established terms before their abbreviations are used. More discussions on how the subculture of 25 years has affected asexual reproduction is needed [3].

Major Comments

References are not up to date, especially in asexual reproduction. Two [4, 5] are closely related to the authors’ manuscript. Please explain and discuss the differences between this study and Pucker et al. 2016. Methodologies should be described in details. For example, “All sequencing data generated in this study were submitted to ENA as part of the study PRJEB33589 and sample ERS3588070.” Based on the information given, no sequence by Pucker et al, PRJEB33589, and ERS3588070 would appear. Page 3, Line 133. What does the north of chromosome 2 mean? Please refer to https://ghr.nlm.nih.gov/primer/howgeneswork/genelocation.

Page 4, 137-141. Please use the term consistently, haploid, diploid, triploid, pentaploid; or 1n, 2n, 3n, 5n.

Page 4, line 158-159, What did this sentence mean? Explain and describe figures and supplementary materials, instead of short sentences and brief phrases. Explain any abbreviations and symbols used. For example, ENA as European Nucleotide Archive https://www.ebi.ac.uk/ena. Page 4, Line 155-158. “This frequency is lower than previously reported values of 1 variant in 200-400 bp for the comparison of thaliana accessions [6,9]. In agreement with previous findings [6,9], the genome-wide distribution showed increased variant frequencies around the centromeres (Figure 2)”. Why lower? Please discuss. Page 6, 207-208.

Minor Comments

References were not well managed; some by reference manager and the others by hand. For example, Page 3, line 115. Need proof reading. For example, Page 3, Line 96-97. “Completeley, deleted regions were identified…”

References Cited

Pucker, B., D. Holtgräwe, T. Rosleff Sörensen, R. Stracke, P. Viehöver and B. Weisshaar, A De Novo Genome Sequence Assembly of the Arabidopsis thaliana Accession Niederzenz-1 Displays Presence/Absence Variation and Strong Synteny. PLOS ONE, 2016. 11(10): p. e0164321. Pucker, B., C. Rückert, R. Stracke, P. Viehöver, J. Kalinowski and B. Weisshaar, 25 years of propagation in suspension cell culture results in substantial alterations of the Arabidopsis thaliana genome. bioRxiv, 2019: p. 710624. Kaas, C.S., C. Kristensen, M.J. Betenbaugh and M.R. Andersen, Sequencing the CHO DXB11 genome reveals regional variations in genomic stability and haploidy. BMC Genomics, 2015. 16(1): p. 160. Schranz, M.E., L. Kantama, H. De Jong and T. Mitchell-Olds, Asexual reproduction in a close relative of Arabidopsis: a genetic investigation of apomixis in Boechera (Brassicaceae). New Phytologist, 2006. 171(2): p. 425-438. Lovell, J.T., R.J. Williamson, S.I. Wright, J.K. McKay and T.F. Sharbel, Mutation Accumulation in an Asexual Relative of Arabidopsis. PLOS Genetics, 2017. 13(1): p. e1006550.

Author Response

Dear Reviewer,

Thank you very much for your time, efforts, and constructive suggestions. Please see below a detailed response to your points.

Best wishes on behalf of all authors,
Boas Pucker

1) References are not up to date, especially in asexual reproduction.

We included the suggested references and additional recent publications in the discussion. Here is an incomplete list of new references:

Bartels et al., 2018: Dynamic DNA Methylation in Plant Growth and Development. https://www.ncbi.nlm.nih.gov/pmc/articles/PMC6073778/

Thiebaut et al., 2019: A Role for Epigenetic Regulation in the Adaptation and Stress Responses of Non-model Plants.
https://www.ncbi.nlm.nih.gov/pmc/articles/PMC6405435/
Sanchez-Munoz et al., 2019: Genomic methylation in plant cell cultures: A barrier to the development of commercial long-term biofactories.
https://onlinelibrary.wiley.com/doi/full/10.1002/elsc.201900024

Fossi et al., 2019: Regeneration of Solanum tuberosum Plants from Protoplasts Induces Widespread Genome Instability.
http://www.plantphysiol.org/content/180/1/78

2) Two [4, 5] are closely related to the authors’ manuscript. Please explain and discuss the differences between this study and Pucker et al. 2016.

Although samples of the same species (Arabidopsis thaliana) were sequenced, objectives and results are completely different between both studies. In our study in 2016, we assembled the genome sequence of the A. thaliana accession Nd-1 and compared it against Col-0. Here, we are not investigation differences between accessions, but look at how the genome of a cell changes under suspension cell culture conditions. Results of all three studies are discussed at several points throughout the now revised “results&discussion” section, for example line153-line158, line196-line197, and line225-line226.

3) Methodologies should be described in details. For example, “All sequencing data generated in this study were submitted to ENA as part of the study PRJEB33589 and sample ERS3588070.” Based on the information given, no sequence by Pucker et al, PRJEB33589, and ERS3588070 would appear.

These data sets are submitted, but not released yet. All submitted data are under an embargo until publication of the corresponding article. These data sets will automatically become publicly available at that point.

4) Page 4, Line 155-158. “This frequency is lower than previously reported values of 1 variant in 200-400 bp for the comparison of thaliana accessions [6,9]. In agreement with previous findings [6,9], the genome-wide distribution showed increased variant frequencies around the centromeres (Figure 2)”. Why lower? Please discuss.

We added explanations for the observed difference in variant frequency: "Variants in At7 are the result of spontaneous mutations which accumulate in the asexually reproducing At7 cells according to Muller’s Ratchet [47] as previously reported for apomicts [48,49]." (line187-line189).

5) Page 6, 207-208. References were not well managed; some by reference manager and the others by hand. For example, Page 3, line 115. Need proof reading.

Thank you for pointing it out. We corrected it.

6) Page 3, Line 96-97. “Completeley, deleted regions were identified…”

Thank you for pointing it out. We corrected it.